# Determinants of maternal influenza vaccination in the context of low- and middle-income countries: A systematic review

Shrish Raut[1‡], Aditi Apte[2‡]*, Manikandan Srinivasan[3], Nonita Dudeja[4], Girish Dayma[1], Bireshwar Sinha[5], Ashish Bavdekar[6]

1 PRERNA Young Investigator, KEM Hospital Research Centre, Pune, India, 2 PRERNA Young Scientist, KEM Hospital Research Centre, Pune, India, 3 PRERNA Young Investigator, Christian Medical College, Vellore, India, 4 PRERNA Young Investigator, Centre for Health Research and Development, Society for Applied Studies, New Delhi, India, 5 PRERNA Young Scientist, Centre for Health Research and Development, Society for Applied Studies, New Delhi, India, 6 Associate Professor, Department of Pediatrics; Consultant, Pediatric Research & Vadu Rural Health Program, KEM Hospital Research Centre, Pune, India

‡ SR and AA are joint first authorship on this work.
* aditi.apte@kemhrcvadu.org

**Data Availability Statement:** The Supporting information files are available at figshare (doi:10.6084/m9.figshare.16757044).

## Abstract

### Background

Pregnancy and early infancy are considered to be the vulnerable phases for severe influenza infection causing morbidity and mortality. Despite WHO recommendations, influenza is not included in the immunization programs of many low- and middle-income countries. This systematic review is aimed at identifying barriers and facilitators for maternal influenza vaccination amongst the perinatal women and their health care providers in low- and middle-income countries.

### Methods

We selected 11 studies from the 1669 records identified from PubMed, CABI, EMBASE and Global Health databases. Studies related to both pandemic and routine influenza vaccination and studies conducted amongst women in the antenatal as well as postnatal period were included. Both qualitative, quantitative, cross-sectional and interventional studies were included.

### Results

Knowledge about influenza disease, perception of the disease severity during pregnancy and risk to the foetus/newborn and perceived benefits of influenza vaccination during pregnancy were associated with increased uptake of influenza vaccination during pregnancy. Recommendation by health care provider, vaccination in previous pregnancy and availability of vaccine in public health system facilitated vaccine uptake. High parity, higher education, vaccination in the later months of pregnancy, less than 4 antenatal visits, concerns about vaccine safety and negative publicity in media were identified as barriers for influenza

**Funding:** The author(s) received no specific funding for this work.

**Competing interests:** The authors have declared that no competing interests exist.

vaccination. Lack of government recommendation, concerns about safety and effectiveness and distrust in manufacturer were the barriers for the healthcare providers to recommend vaccination.

## Conclusion

While availability of influenza vaccine in public health system can be a key to the success of vaccine implementation program, increasing the awareness about need and benefits of maternal influenza vaccination amongst pregnant women as well as their health care providers is crucial to improve the acceptance of maternal influenza vaccination in low and middle-income countries.

## Background

Pregnant women and neonates are known to be vulnerable to severe influenza disease complications, and death [1–3]. Globally, the influenza-related hospitalisation rate in pregnant women is estimated to be 42.1% (interquartile range (IQR), 22.5–60.4%) and among them around 8% (IQR, 5.9–12.7%) have severe disease that results in intensive care admission or death [4]. Review of eight Indian studies have reported maternal mortality rate of 25–75% in pregnant women with influenza [5]. Influenza infection during pregnancy is also associated with poor birth outcomes viz. foetal loss (abortion or still birth), preterm birth and low birthweight [1, 5–7]. The Strategic Advisory Group from the World Health Organization (WHO-SAGE) universally recommends vaccination of pregnant women against influenza [8]. The licensed inactivated trivalent influenza vaccine (IIV3) is recommended for use in any trimester of pregnancy, to protect the mother as well as her newborn till 6 months of age. A systematic review from tropical and subtropical countries demonstrated that influenza vaccination in pregnant women can prevent laboratory-confirmed influenza in pregnant women (50%) as well as in their infants <6 months (49–63%) [9]. However, maternal influenza vaccination is not included in the immunisation programs of many low and middle-income countries (LMIC) and coverage of influenza vaccine remains low in pregnant women globally, especially in resource-constrained settings in LMICs [10–12].

Public health decision-making related to maternal influenza vaccination is challenging as health priorities vary across countries, and the comprehensive evidence on disease burden is lacking for LMICs [5]. Maternal influenza vaccine coverage is influenced by several stakeholder-linked factors which may be manufacturer-related, heath care provider (HCP)- related or pregnant women-related. Several barriers have been reported globally, to maternal influenza vaccination that include lack of awareness about influenza disease among major stakeholders (mothers, health care workers, doctors), vaccine hesitancy, technical challenges in provision of influenza vaccination services and socio-cultural issues [10, 13]. Lack of HCP-endorsement on influenza vaccination, hesitancy of the HCPs to vaccinate pregnant women [14], financial barriers, and lack of clear national recommendations are common obstacles to antenatal influenza vaccination reported in the global literature [15, 16]. Additionally, safety concerns for the foetus, lack of awareness regarding the severity and burden of influenza, and poor knowledge of the benefits of vaccination are identified factors for poor vaccine uptake in pregnancy in global literature reviews [17]. However, there is lack of systematic evidence on the uptake of maternal influenza vaccine among all stakeholders from LMICs.

Given that pregnant women are one of the critical target groups, there is a need to research the local and contextual factors for poor uptake of maternal influenza vaccine in LMICs. This systematic review was planned to synthesize evidence about barriers and facilitators for maternal influenza vaccination amongst HCPs and pregnant women in LMICs.

## Methods

The review has been registered at PROSPERO registry for systematic reviews (https://www.crd.york.ac.uk/prospero/display_record.php?ID=CRD42021243363).

### Inclusion and exclusion criteria

Studies from LMICs assessing the uptake of maternal influenza vaccination or knowledge, attitude and perception regarding influenza vaccination in pregnant women or health care providers were included in the review. We included studies published until Dec 31, 2020 amongst women in the antenatal as well as postnatal period. Studies related to both pandemic and routine influenza vaccination were included. Both qualitative as well as quantitative, cross-sectional or interventional studies were included. Studies from high income countries, those assessing efficacy of maternal influenza vaccination or studies on uptake of influenza vaccination amongst population other than pregnant women were excluded from the review.

### Literature search strategy

The search strategy was finalized based on a pilot exercise. During the pilot exercise, multiple search terms were identified for the population (pregnant women, health care providers), exposure (influenza, vaccination) and outcome (vaccine acceptance and its determinants). The final search strategy was developed through an iterative process and discussions with all authors. The search strategy utilised combinations of MeSH and non-MeSH terms. The detailed search strategy has been depicted in Table 1.

The developed search strategy was used to search PubMed, CABI, EMBASE and Global Health databases. The search had no date range included and all studies in English language were included in the search. Studies from LMIC were identified based on the status of the country at the time of publication as per the World Bank definition [18]. The initial search was performed by one author (SR). The results were then filed together using Mendeley Referencing Software and duplicate articles were removed. The titles and abstracts of all the remaining

**Table 1. Search criteria.**

| No | Search criteria |
|----|-----------------|
| #1 | (((((((((Pregnant[Title/Abstract]) OR (Maternal[Title/Abstract])) OR (postnatal[Title/Abstract])) OR (pregnancy[Title/Abstract])) OR (lactating[Title/Abstract])) OR (expectant[Title/Abstract])) OR (mother[Title/Abstract])) OR (antenatal[Title/Abstract])) OR (pueperal[Title/Abstract]) |
| #2 | (((((((Healthcare[Title/Abstract] AND provider[Title/Abstract]) OR (doctor[Title/Abstract])) OR (physician[Title/Abstract])) OR (obstetrician[Title/Abstract])) OR (gynaecologist[Title/Abstract] OR gynecologist[Title/Abstract])) OR (nurse[Title/Abstract])) OR (practitioner[Title/Abstract])) OR (caregiver[Title/Abstract]) |
| #3 | (vaccin*[Title/Abstract]) OR (immun*[Title/Abstract]) |
| #4 | ((((((((((((accept*[Title/Abstract]) OR (uptake[Title/Abstract])) OR (predictor[Title/Abstract])) OR (facilitator[Title/Abstract])) OR (determinant[Title/Abstract])) OR (barrier[Title/Abstract])) OR (factor[Title/Abstract])) OR (recommendation[Title/Abstract])) OR (knowledge[Title/Abstract])) OR (attitude[Title/Abstract])) OR (practice[Title/Abstract])) OR (willlingness[Title/Abstract]) |
| #5 | ((influenza[MeSH Major Topic]) OR (flu[Title/Abstract])) OR (H1N1[Title/Abstract]) |
| #6 | #1 OR #2 |
| #7 | #6 AND #3 AND #4 AND #5 |

studies were screened by two reviewers independently (SR and MS). Full texts of all the selected articles were independently screened by two authors using predefined inclusion criteria for the review (SR and GD). Any disagreements were settled through discussion with a third author (AA). Apart from the published literature, workshop reports or conference proceedings were searched through the given databases and Google as well as through reference lists of the published papers on this topic.

## Data extraction and analysis

A data extraction template in excel was prepared to extract data from the selected full texts. The template was designed to capture following details of the studies: study type, year, setting, sample size, country, type of vaccine, determinants of vaccine uptake, barriers perceived by pregnant women and health care providers, limitations of the study. The data was extracted by one author (SR) and was verified by a second author (MS).

Quality assessment was done for all the included studies independently by two authors (SR and ND). Quality assessment for observational studies was assessed using New Castle Ottawa Scale [19, 20], while that for the qualitative studies was done using a quality assessment tool developed by Hawker et al [21].

The quantitative data on determinants of uptake of maternal influenza vaccine was summarised into tables using estimates on proportions and odds ratios for various factors. Combined frequency tables for perceived barriers were prepared to summarise data from qualitative and quantitative studies.

## Results

A total 2434 records were identified from the four databases, of which 1669 title/abstracts were screened after removing 765 duplicate records. A total of 323 full texts were screened of which 11 records were included based on the predefined inclusion criteria. Fig 1 shows PRISMA flow chart for selection of studies with reasons for exclusion.

### Overview of included studies

More than half of the studies (7 out of 11) were cross-sectional surveys, whereas the remaining four studies used qualitative or mixed methods approach. The studies were conducted between 2010 to 2017. There were four studies from Africa (Malawi [22], Ethiopia, Ghana, Uganda, Laos [23], Gambia [24], Ivory Coast [25]), four from the Americas (El Salvador, Peru [26, 27], Nicaragua [28, 29]), two from Eastern Mediterranean region (Pakistan [30], Morocco [31]) and one from South-East Asia (India) [32]. Ten studies included pregnant or recently pregnant women and three studies included HCPs as respondents for the study. The studies included were conducted either in urban and rural setting (3 studies) or in urban settings (8 studies). In three of included studies [26, 28, 29], influenza vaccine was available to the pregnant women and factors associated with actual vaccine uptake were assessed. In all the other studies, factors associated with willingness to receive maternal influenza vaccine was assessed through qualitative or quantitative methods. Detailed study characteristics are given in Table 2.

### Quality assessment

Nine quantitative studies were assessed using New Castle Ottawa scale, of which, five had low risk-of-bias; four studies had high risk-of-bias due to lack of justification for sample size and use of non-validated study tool [22, 28, 30, 32]. Two qualitative studies included were assessed to have low [23] and moderate [31] quality using quality assessment tool developed by Hawker

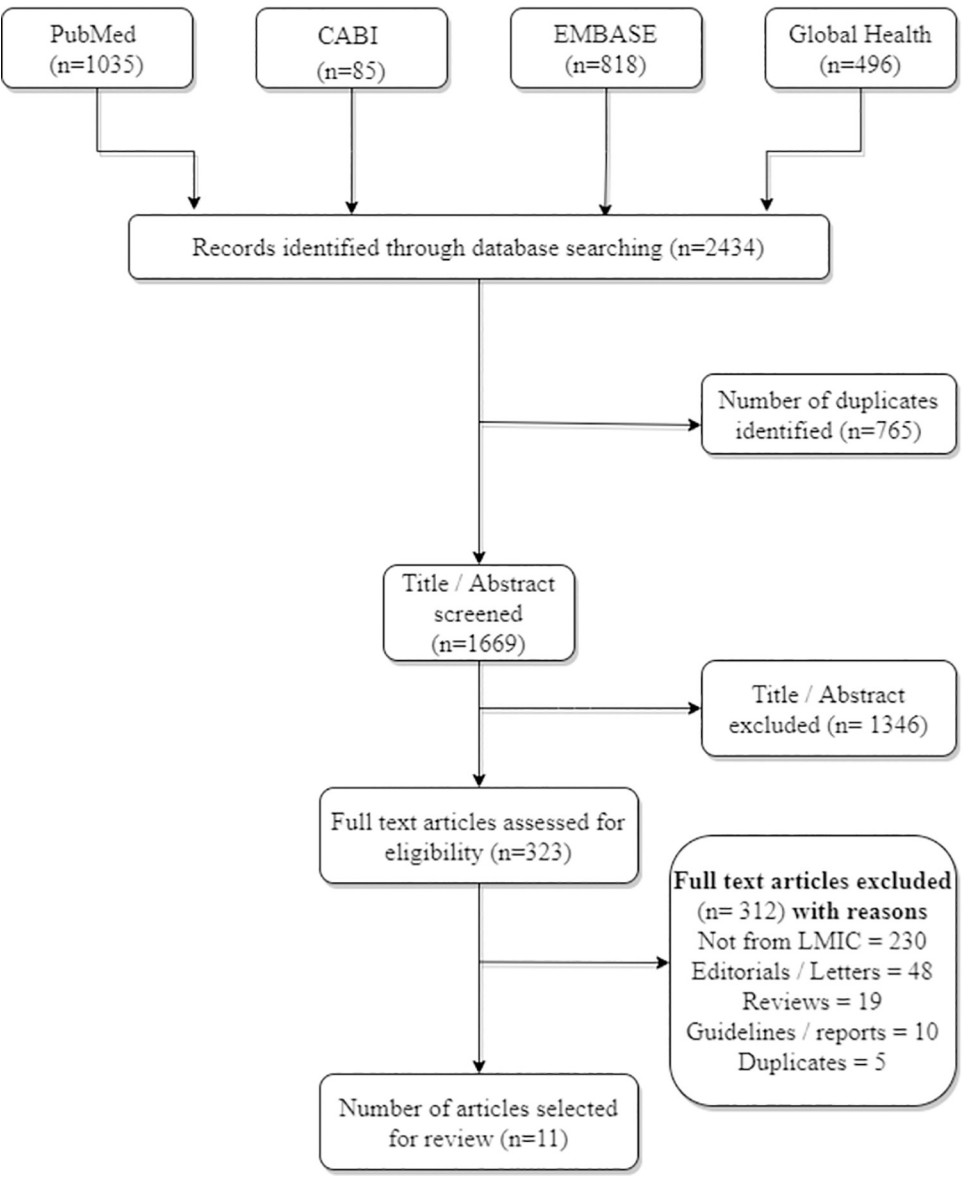

**Fig 1. PRISMA flow chart.**

et al [Table 2]. The detailed quality assessment of included studies is provided in S1 and S2 Tables.

## Vaccine acceptance or vaccine uptake

Outcomes in the studies included overall vaccine uptake as well as vaccine acceptance (patient agreement to vaccine assuming it was offered) [Table 3]. Studies from Peru [26] and Nicaragua [28, 29] reported influenza vaccination rate of 19–28% and 42–71% respectively amongst pregnant women. Amongst studies assessing willingness to receive maternal influenza vaccine acceptance rate of 45%, 87% and 98.5% was reported from Ivory coast [25], Pakistan [30], Gambia [24]. In the study conducted by Arriola *et al* in Nicaragua [29], influenza vaccine was recommended in pregnancy by 89% HCPs [Table 2].

**Table 2. Study characteristics.**

| Study (Author and year) | Country | Study duration of (month and year) | Study design | Study Setting | Type of study population and settings | Influenza Vaccination / acceptance rate (%) | Quality |
|---|---|---|---|---|---|---|---|
| Reinders et al 2019 [26] | Peru | July and August 2016 | Cross sectional | Urban | Mothers of children <5 years (n = 624) | 28% vaccinated | Low risk of bias$ |
| | | | | | Pregnant women (n = 54) | 19% vaccinated | |
| Fleming et al 2019 [(22] | Malawi | 2015 | Mixed method | Urban + Rural | Pregnant or recently pregnant women (n = 274) and others* | NA | Very high risk of bias$ |
| Arriola et al 2018 [29] | Nicaragua | June and August 2016 | Cross sectional | Urban | Pregnant women (n = 1303) | 42% vaccinated | Low risk of bias$ |
| | | | | | Health Care Providers (n = 600) | 89% recommended vaccine | |
| Top et al 2018 [23] | Ethiopia, Ghana, Uganda, and Laos | September October 2015 | Qualitative | Urban | Health Care Providers (n = 141) | NA | Low quality # |
| Armitage et al 2018 [24] | Gambia | August and September 2017 | Cross sectional | Urban | Non pregnant women (n = 454) Vaccinated 150 and control 304 | 98.5% willing to be vaccinated in pregnancy | Low risk of bias$ |
| Fleming et al 2018 [27] | El Salvador | 2015–2016 | Mixed method | Urban | Pregnant or recently pregnant women (n = 117) and others* | NA | Low risk of bias$ |
| Arriola et al 2016 [28] | Nicaragua | October and December 2013 | Cross Sectional | Urban | Pregnant women (n = 1807) | 71% vaccinated | Very high risk of bias$ |
| Khan et al 2015 [30] | Pakistan | May to August 2013 | Cross Sectional | Urban | Pregnant women (n = 274) | 87% willing to be vaccinated | High risk of bias$ |
| Koul et al 2014 [32] | India | October 2012, and April 2013. | Cross Sectional | Urban + Rural | Pregnant women (n = 1000) | None received vaccine | Very high risk of bias$ |
| | | | | | Health Care Providers (n = 90) | None recommended vaccine | |
| Lohiniva et al 2014 [31] | Morocco | October 2010 | Qualitative | Urban + Rural | Pregnant women (n = 123) Vaccinated 67 and Unvaccinated 56 | NA | Moderate quality # |
| Kouassi et al 2012 [25] | Ivory coast | February 2010 | Cross Sectional | Urban | Pregnant women (n = 411) | 45% intended to be vaccinated | Low risk of bias$ |

$ Quality assessment of crosssectional and mixed method studies was done by Newcastle Ottawa Scale

# Quality assessment for qualitative studies done using Hawker et al method [43]

* family members, community leaders, public health program managers, non-governmental partners, and policy makers.

## Demographic and clinical factors associated with uptake of influenza vaccination by pregnant women [Table 3]

Women with higher education were 36–42% less likely to accept influenza vaccination during pregnancy as compared to women with primary or less than primary education [26]. Women having three or more children were less likely to take the vaccine as compared to women with one child (76% vs. 98%, OR = 0.08) [30]. Women were more willing to receive influenza vaccine in first or second trimester against third trimester (95% vs. 85%, OR = 0.03) [30]. Also, those who received influenza vaccine in the previous pregnancy were more likely to receive it in the present pregnancy [29]. Women with high-risk obstetric condition were less likely to be vaccinated against influenza (36% vs. 45% [29], whereas those with existing medical conditions were 4 times more likely to be vaccinated (26). Demographic factors such as age, socioeconomic status, marital status or employment status were not found to influence decision making regarding vaccine uptake.

Non-availability of time and need for permission from husbands or one of the family members was perceived as a barrier to receive influenza vaccination in a few studies [Table 4].

**Table 3. Factors associated with uptake and acceptance of maternal influenza vaccine amongst pregnant women and/or health care providers.**

| Factors | References | Proportion amongst vaccinated / unvaccinated women or women with or without intention to receive the vaccine | Odds ratio or risk ratios associated with significant likelihood of vaccination/acceptance of vaccination | Phenomenon addressed |
|---|---|---|---|---|
| **Demographic and clinical factors** | | | | |
| Women with high school or technical education | [26] | – | High school education—0.64 (95% CI 0.49–0.83) Technical education—0.58 (95% CI 0.43–0.79) Ref: Primary education or less | Vaccine uptake |
| Having more than three children | [30] | 76% in women with three or more children vaccinated vs. 98% in women with one child; p = 0.02 | 0.08 (95% CI 0.01–0.63) Ref: Having one child | Vaccine acceptance |
| Vaccination in third trimester of pregnancy | [30] | 85% women in third trimester vaccinated vs 95% among women in 1st or 2nd trimester; p = 0.03 | 0.3 (95% CI 0.1–0.87) Ref: Vaccination in 1st or 2nd trimester | Vaccine acceptance |
| Presence of high-risk obstetric condition | [29] | 36% women with HROC vaccinated vs 45% women without HROC, p = 0.002 | – | Vaccine uptake |
| Pre-existing medical condition | [26] | – | 4.20 (95% CI: 2.03–8.70) Ref: No pre-existing medical condition | Vaccine uptake |
| Receipt of flu vaccine in previous pregnancy | [29] | 32% vaccinated women vs. 14% unvaccinated women reported receipt of flu vaccine in previous pregnancy, p<0.001 | — | Vaccine uptake |
| Four or more antenatal visits | [28] | | 2.58(95% CI 1.15, 5.81) Ref: One antenatal visit | Vaccine uptake |
| **Knowledge about disease, perceived risk of illness and protection offered by the vaccine in pregnant women** | | | | |
| Knowledge about influenza disease | [29] | 98% vaccinated vs. 75% unvaccinated women knew about flu, p<0.001 | — | Vaccine uptake |
| | [30] | 94.1% women with and 45.7% without the intention to get vaccinated knew about flu, p<0.0001 | 24.28(95% CI 9.88–59.68) Ref: No knowledge of the disease | Vaccine acceptance |
| Perceived risk of influenza disease during pregnancy | [29] | 88% vaccinated vs.68% not vaccinated perceived the risk, p<0.001 | | Vaccine uptake |
| | [30] | 45.8% with and 25.7% without intention to get vaccinated perceived the risk, p = 0.03 | 2.38 (95%CI 1.07–5.32) Ref: No perceived risk of influenza | Vaccine acceptance |
| Perceived risk of influenza to infants | [30] | 76.6% with and 50% without intention to get vaccinated perceived the risk, p = 0.0004 | 3.80(95% CI 1.81–7.98) Ref: No perceived risk of influenza | Vaccine acceptance |
| Need for influenza vaccination during pregnancy | [30] | 93%vs7%,p<0.0001 | ——————————— | Vaccine acceptance |
| Perceived safety of influenza vaccine for the mother | [29] | 95% vaccinated vs.77% unvaccinated women perceived the safety, p<0.001 | | Vaccine uptake |
| | [30] | 77.4% with and 28.5% without intention to receive the vaccine perceived the safety<0.0001 | 10.09(95%CI 10.09 4.50–22.63) Ref: Vaccine not perceived safe | Vaccine acceptance |
| Perceived effectiveness of influenza vaccine for the mother | [29] | 95% vaccinated vs.77% unvaccinated women perceived the benefit, p<0.001 | | Vaccine uptake |
| | [30] | 80.7% with and 37.1% without the intention to receive vaccinated perceived the benefit, p<0.0001 | 8.43 (95%CI 3.88–18.31) Ref: Vaccine not perceived efficacious | Vaccine acceptance |
| Perceived protection for influenza for infant | [30] | 85.8% with and 40% without the intention to receive vaccinated perceived the benefit, p<0.0001 | 9.45 (95% CI 4.33–20.62) Ref: No perceived protection | Vaccine acceptance |
| **Health provider related factors** | | | | |
| Recommendation from physicians for flu vaccine | [29] | 81% vaccinated vs.5% unvaccinated women had received a recommendation, p<0.001 | 74.11 (95% CI 36.63–149.94) | Vaccine uptake |
| | [30] | 82% with and 24% without the intention to receive vaccine had received a recommendation, p<0.01 | 2.47 (95% CI 1.16–5.28) | Vaccine acceptance |

*(Continued)*

**Table 3.** (Continued)

| Factors | References | Proportion amongst vaccinated / unvaccinated women or women with or without intention to receive the vaccine | Odds ratio or risk ratios associated with significant likelihood of vaccination/acceptance of vaccination | Phenomenon addressed |
|---|---|---|---|---|
| | [28] | | 14.22 (95% CI 10.45–19.33) Ref: No recommendation from HCP for flu vaccine | Vaccine uptake |
| Recommendation from HCP for any vaccine during pregnancy | [30] | 75.7% with and 60% without intention to receive the vaccine has received a recommendation, p = 0.02 | 2.55(95% CI1.18–5.48) Ref: No recommendation for HCP | Vaccine acceptance |
| Received offer for influenza vaccination from health care provider | [29] | 95% of the women who received offer vaccinated vs. 5% who did not receive, p<0.01 | 15.69(95%CI 7.45–33.03) Ref: Vaccination not offered. | Vaccine uptake |
| Belief that physicians are reliable source of vaccine information | [30] | – | 7.55(95%CI 2.06–27.67) Ref: Physicians are not a reliable source of information | Vaccine acceptance |
| **Health system related factors** | | | | |
| Vaccination in private clinic set up | [25] | – | 0.19(95%CI 0.05–0.76) Ref: Vaccination in public health set up | Vaccine acceptance |

HROC- High risk obstetric condition; vaccine uptake means actual receipt of vaccine, vaccine acceptance means willingness or intent to get vaccinated

## Factors related to knowledge of disease and vaccine-related factors

Five studies used health belief constructs to assess the predictors of vaccine acceptance. Knowledge about influenza disease amongst pregnant women was associated with increased likelihood of receiving the vaccine during pregnancy [OR = 24.28]. Perceived risk of influenza disease during pregnancy and to the newborn were associated with increased acceptance for influenza vaccination during pregnancy [OR = 2.38 and 3.8 respectively] [29, 30]. Further, perceived need for influenza vaccination during pregnancy was associated increased acceptance for the vaccine [93% vs. 7%]. Women who perceived the vaccine to be safe and effective were 8–10 times more likely to receive vaccine as compared to those who did not perceive the benefit [29, 30] [Table 3].

Concern about safety of the vaccine to self was identified as a barrier in seven studies and was reported by up to 52% women [25, 26, 29, 30, 31]. Safety concern to the unborn child was identified as a barrier by 15% women in a study conducted by Khan et al [30]. Distrust about the vaccine was identified as a barrier in five studies [25, 26, 29, 30, 31] or a perception that the vaccine is not needed was identified as a barrier for vaccination in one study [26]. Lack of awareness about maternal influenza vaccination was perceived a barrier in four studies [25, 28, 29] and by up to 55% respondents [Table 4].

## Factors related to healthcare providers and health system

The odds of receiving influenza vaccine were reported 2.5 to 74 times higher in pregnant women who received a recommendation from an HCP as compared to those who did not receive any recommendation [28–30]. In fact, recommendation for any vaccine during pregnancy was associated with 2.5 times increased acceptance of influenza vaccine during pregnancy [30]. Further, women who were offered influenza vaccine by their HCPs were even more likely to receive the vaccine as compared to those who merely received a recommendation (95%vs.81%, p<0.01) [29]. Women who trusted their care-providers during pregnancy were seven times more likely to receive influenza vaccine [30]. Women who received antenatal care in private set up were less likely to receive influenza vaccine during pregnancy as compared to public health set up [OR = 0.19] [25]. Women with four or more antenatal visits were twice likely get vaccinated for influenza than those with less than four antenatal visits [28] [Table 3].

**Table 4. Barriers to maternal influenza vaccination as perceived by pregnant women and health care providers[*].**

| Barriers to influenza vaccination perceived by pregnant women | *Reinders et al 2019* [26] | *Fleming et al 2019* [22] | *Arriola et al 2018* [29] | *Top et al 2018* [23] | *Armitage et al 2018* [24] | *Fleming et al 2018* [27] | *Arriola et al 2016* [28] | *Khan et al 2015* [43] | *Koul et al 2014* [32] | *Lohiniva et al 2014* [31] | *Kouassi et al 2012* [25] |
|---|---|---|---|---|---|---|---|---|---|---|---|
| Safety concern to self | ✔(52%) | ✔ | ✔(1%) | | | ✔(14%) | | ✔(17%) | | ✔ | ✔(10%) |
| Safety concerns to unborn child | | | | | ✔(50%) | | | ✔(15%) | | | |
| Distrust for vaccine | ✔(11%) | | | | | | ✔ | ✔(11%) | | ✔ | ✔(12.5%) |
| Non availability of time | ✔(14%) | | ✔(0.2% | | | ✔(3.5%) | | | | | |
| Unaware of vaccine & / or its necessity | ✔(55%) | | ✔(41%) | | | | ✔(44%) | | | | ✔(45%) |
| Need for permission from husband / household member | | | | | | ✔(27.6%) | | ✔(30%) | | | |
| Belief that vaccine not needed | | | ✔(3%) | | | | | | | | |
| Vaccine not been offered by HCP | | | ✔(56%) | | | ✔(3.5%) | ✔ (10%) | | ✔ (100%) | | |
| Non-availability | ✔(5%) | ✔ | ✔(3.7%) | | | | ✔(2%) | | | ✔ | |
| Non-accessibility | | | | | | ✔ | | | | | |
| Negative or no counselling by HCPs | | ✔ | ✔(0.9%) | | | | | | | | |
| Negative publicity by media | ✔(9%) | | | | | ✔ | | | | | |
| Combining antenatal services with vaccine | | ✔ | | | | | | | | | |
| Lack of respect/Poor treatment by HCP | | ✔ | | | | | | | | | |
| Unknown reason | ✔(2%) | | ✔(34%) | | | | | | | | |
| **Barriers to influenza vaccination as perceived by health care providers** | *Reinders et al 2019* | *Fleming et al 2019* | *Arriola et al 2018* | *Top et al 2018* | *Armitage et al 2018* | *Fleming et al 2018* | *Arriola et al 2016* | *Khan et al 2015* | *Koul et al 2014* | *Lohiniva et al 2014* | *Kouassi et al 2012* |
| Not in government / public health policy | | | | ✔ | | | | | | | |
| Safety concerns about the vaccine | | | ✔(2%) | ✔ | | | | | ✔ | | |
| Distrust about vaccine / manufacturer | | | | ✔ | | | | | ✔ | | |
| Short Shelf life of influenza vaccine | | ✔ | | | | | | | | | |
| Lack of health information system to track vaccination coverage | | ✔ | | | | | | | | | |
| Lack of social harmony (internal conflict/gang activity) | | | | | | ✔ | | | | | |

[*]The table includes results from both qualitative and quantitative studies. Proportions in % are provided from quantitative studies wherever available indicating the percentage of the study participants who reported the barriers related to influenza vaccination; HCP- Health care providers.

On the other hand, four studies reported that women did not receive influenza vaccination because it was not offered to them by their HCPs [27–29, 32]. Negative counselling by HCPs was identified as a barrier for maternal influenza vaccination in two studies [22, 29]. Lack of

respect by the HCP towards confidentiality of study participants or poor treatment by HCP were identified as barriers in one of the studies [22].

Similar to pregnant women, concerns over safety of influenza vaccine during pregnancy or distrust about the vaccine manufacturer were barriers identified amongst HCPs in three studies [23, 29, 32]. In addition, short shelf life of the product and lack of essential safety information and ambiguous nature of product monograms [22] was reported as a barrier in the Malawi study. Lack of availability of influenza vaccine in government policy was reported as an important barrier amongst HCPs [23]. Presence of health information system was perceived necessary in Malawi study to keep track of vaccination coverage when pregnant women visit multiple health centres for antenatal check-ups and absence of such system was perceived as an operational challenge [22]. Presence of criminal gang activity was found to a barrier in El Salvador study as pregnant women had limited access to health services in these insecure areas and needed permissions from the gang leaders to attend clinics [27] [Table 4].

### Influence of family, community and media

Apart from health care providers, community health workers and friends /neighbours were identified as important sources of information regarding influenza vaccination by 46% and 34% women respectively [31].

Husbands, family members, friends especially non-medical ones, neighbours and relatives were found to influence the decision-making process with husband being the most influential person among them. Recommendation by governmental bodies was considered as one of the reliable sources of vaccine information by 33% of study participants and was significantly associated with increased acceptance of influenza vaccine [OR = 3.52] [30]. Discussions with neighbours and friends in some women led to reduced acceptance of vaccination. These discussions were often based on instances about complications and side effects affecting those who had been vaccinated. None-the-less, they were considered trusted advisors [31].

Among the media sources, television (69–72%), radio (32–44%) and text messages received on mobile phone (75–83%) were found to be positive influencers for influenza vaccination during pregnancy [25, 26, 31].

### Discussion

This systematic review addresses key determinants which facilitate pregnant women to consider influenza vaccination in LMIC settings, and the potential barriers to influenza vaccination uptake both from the perspective of pregnant women as well as healthcare providers. The review has included studies from South Asia, Africa, America and Eastern Mediterranean regions and thus presents findings from diverse geographical and sociocultural contexts. We found that in the different studies, influenza vaccination rate among pregnant women varied between 19 to 71% which is comparable to coverage rates amongst pregnant women some high-income countries [33]. However, the coverage for influenza vaccination was lower during pregnancy than other populations (e.g., children and elderly) in Peru and Nicaragua [34]. These findings are similar to other global data indicating low uptake of influenza vaccine during pregnancy [34]. The vaccine uptake ranged between 45 to 99%, highest in Gambia [24] and lowest in Ivory Cost [25] region.

Several constructs of health belief model were found to influence the decision-making regarding influenza vaccination during pregnancy [35]. Knowledge about influenza disease, perception of the disease severity during pregnancy and risk to the foetus/newborn and perceived benefits of influenza vaccination during pregnancy were associated with increased acceptance of influenza vaccination during pregnancy. Cues to action, especially,

recommendation by health care provider or government authorities and history of vaccination in previous pregnancy were strong influencers of vaccine uptake. On the other hand, lack of perception of disease severity or need for vaccination, safety concerns about the vaccine for self or to the unborn child, distrust for vaccine were the perceived barriers. Higher education without specific knowledge on the disease, lack of clear health information, non-availability of vaccines and negative publicity by media were some other barriers reported for maternal influenza vaccination. Our findings are similar to those reported recently by Yuen et al in their global systematic review on determinants of influenza vaccine uptake during pregnancy [36]. Improved health literacy was a found to be a facilitator, which has been reported earlier from high income settings [36]. Overall, these findings indicate that improved public health education about risk of influenza during pregnancy and importance of maternal influenza vaccination can potentially increase the uptake of vaccines in LMIC settings. Buchy *et al* in their expert commentary have also highlighted the problem of low uptake of maternal influenza vaccine globally due to ineffective communication with the pregnant women about the risks and benefits of influenza vaccination during pregnancy [33].

High parity i.e., having three or more children was reported to be a barrier for maternal vaccination. Earlier literature also reports that primipara women are more willing for influenza vaccination [17, 37]. Offering influenza vaccines early by first or second trimester of pregnancy, higher (≥4) antenatal visits, absence of high-risk obstetric conditions were associated with higher influenza vaccination rate in pregnant women. Pregnant women who engage with the health system early, tend to have better opportunity to discuss with their care-providers about influenza vaccination or get counselled by the HCPs regarding vaccination, resulting in higher uptake. A higher acceptance for vaccination was noted when vaccines were offered in public health facility, compared to private facilities. This could be due to the subsidized rates at which the vaccine is offered in public health system as compared to private facilities, given that the study population is from resource-constrained LMIC settings. Also, availability in public health system gives more credibility due to the underlying government support, thus increasing the overall acceptance by general public as well as health care providers. Thus, ensuring adequate access to antenatal care and inclusion of maternal influenza vaccine in the government policy can be key facilitators for success in maternal influenza vaccination.

Recommendation by the HCPs has been identified as a key determinant for maternal influenza vaccine uptake in previous global literature [22]. Wong et al in their global systematic review on interventions to increase uptake of influenza vaccination in pregnancy have recommended that clinicians should educate the pregnant women about benefits of influenza vaccination in pregnant women and newborns [38]. Morales *et al* in their recent review on determinants of influenza vaccination in pregnancy have identified recommendation by HCPs for influenza vaccination during pregnancy and their perception of safety and efficacy of influenza vaccine in pregnancy as important determinants [17]. The current review reemphasizes the importance of health care providers as a stakeholder in maternal influenza vaccination in the context of developing world where recommendation by HCPs about influenza vaccination and trust in HCPs were major facilitators. On the other hand, negative counselling by HCPs was reported as a barrier. The potential barriers identified by HCP for vaccination include distrust about influenza vaccine manufacturer, inadequacy of safety information from the manufacturer and lack of recommendation for influenza vaccination governed by the national policy. Uncertainty and fear about the safety and benefits of maternal vaccination amongst the HCPs despite recommendation by health authorities [29] and ineffective communication by the HCPs about risk and benefits of maternal vaccination are known concerns [33]. Thus, improving knowledge of HCP about the safety and effectiveness of maternal influenza

vaccination, addressing their concerns along with recommendation by health policy makers on maternal influenza vaccination can increase the vaccine confidence of HCPs in LMICs.

Overall, our review points towards need for increased preparedness about maternal influenza vaccination amongst pregnant women and their families as well as their care providers amongst LMICs in addition to making the vaccine available in health program. The evidence from Peru study [26] suggests that the vaccine uptake may remain low despite subsidized vaccine program. This highlights the importance of including awareness campaigns for general public and health care providers in order to improve vaccine coverage. This can be achieved through national level public health campaigning about the risk of influenza during pregnancy and benefits of maternal vaccination amongst general public and stakeholders in health system. An example from a middle-income country like Argentina has shown that provision of maternal influenza vaccine free-of-cost through health program can lead to about 95% coverage of maternal vaccination [39]. Considering the success of maternal tetanus vaccination in LMICs, the acceptance of maternal influenza vaccination can be enhanced manifold with the presence of health policy recommendation and availability of influenza vaccine through public health program. Even though WHO has recommended upscaling influenza vaccination in its member states [40], having policy recommendations made at country level as well as recommendation by national advisory bodies in obstetrics is important for convincing HCPs to advocate influenza vaccine to pregnant women. Further, there is lack of research focusing on the policy makers at LMIC settings and disease burden of influenza in pregnancy and active surveillance studies for maternal influenza vaccination in LMIC which will help in establishing maternal influenza vaccination as a priority in public health [41, 42].

This is the first review to our knowledge that throws light on determinants of maternal influenza vaccination in LMIC settings. Important strengths of this review include use of comprehensive search terminology and having carried out article search in four databases. However, this review is not without limitations. Having a limited number of articles, of about 11, being included in our review is a serious limitation for generalizing our findings to pregnant women in whole of LMIC settings. Further, most of our evidence is from surveys conducted among pregnant mothers sampled at selected health facilities of the study area, which again questions the representation of these findings to the concerned study population. Except Nicaragua [28] [29] and Peru [26], none of the countries had maternal influenza vaccination included in the health program. Hence, there was limited information available about the logistic or operational factors related to availability of influenza vaccine for pregnant women in these countries, which can play very important role in the success of implementation. Further, four out of 11 studies had high or very high risk of bias which reduces the confidence in the results from these studies. Although this review included evidence of high-quality studies as well, all these studies were observational in nature, with none being conducted in randomized controlled trial settings.

We have excluded full text articles from high income countries and focused on the studies from LMIC only. This is because the problems faced by LMIC are likely to be different from higher income countries due to differences in the socioeconomic status, literacy and health care access for pregnant women. However, we have discussed the findings from these studies at relevant places in the introduction and discussion. Despite the limitations, the review points towards the important fact that research on influenza vaccination in pregnant women has been a low priority in LMIC setting and highlights the need for more population-based studies to enable policymakers understand the critical determinants of influenza vaccination in their settings.

## Conclusion

Higher educational status, better access to antenatal care, perceived risk of influenza during pregnancy, perceived benefits of influenza vaccination during pregnancy, recommendation by health care providers and inclusion of maternal influenza vaccine in health policy were important facilitators for maternal influenza vaccine uptake in LMIC. Fear of adverse effects, uncertainty about the benefits of vaccination and ineffective health communication regarding the influenza vaccine were barriers identified.

Thus, while availability of influenza vaccine in public health system can be a key to the success of vaccine implementation program, increasing the public health awareness about need and benefits of maternal influenza vaccination amongst pregnant women as well as their health care providers is crucial to improve the acceptance of maternal influenza vaccination in low and middle-income countries.

## Supporting information

**S1 Table. Quality assessment of studies using New Castle Ottawa scale.**
(DOCX)

**S2 Table. Quality assessment of qualitative studies using tool developed by Hawker et al.**
(DOCX)

## Acknowledgments

The Young Investigators and Young Scientists acknowledge the core support provided by the Bill and Melinda Gates Foundation (Grant ID OPP1110191). We thank the technical advisory group of the Platform for Research Excellence Related to National Aims (PRERNA) and faculties at Centre for Health Research and Development, Society for Applied Studies, Delhi; KEM Hospital Research Centre, Pune and Christian Medical College, Vellore for their support and guidance.

## Author Contributions

**Conceptualization:** Aditi Apte, Ashish Bavdekar.

**Data curation:** Shrish Raut, Aditi Apte, Girish Dayma.

**Formal analysis:** Shrish Raut.

**Investigation:** Shrish Raut, Manikandan Srinivasan.

**Methodology:** Shrish Raut, Aditi Apte, Manikandan Srinivasan, Nonita Dudeja, Girish Dayma, Bireshwar Sinha.

**Supervision:** Aditi Apte, Ashish Bavdekar.

**Validation:** Shrish Raut, Aditi Apte, Nonita Dudeja, Girish Dayma, Bireshwar Sinha.

**Visualization:** Shrish Raut.

**Writing – original draft:** Shrish Raut, Aditi Apte, Manikandan Srinivasan.

**Writing – review & editing:** Shrish Raut, Aditi Apte, Manikandan Srinivasan, Nonita Dudeja, Girish Dayma, Bireshwar Sinha, Ashish Bavdekar.

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
