## [Decision Letter · Decision Letter 0]

10 Sep 2021

PONE-D-21-08838Determinants of maternal influenza vaccination in the context of low- and middle-income countries: A systematic reviewPLOS ONE

Dear Dr. Apte,

Thank you for submitting your manuscript to PLOS ONE. After careful consideration, we feel that it has merit but does not fully meet PLOS ONE’s publication criteria as it currently stands. Therefore, we invite you to submit a revised version of the manuscript that addresses the points raised during the review process.

We look forward to receiving your revised manuscript.

Kind regards,

Emily A Hurley, M.P.H., Ph.D.

Academic Editor

PLOS ONE

Additional Editor Comments (if provided):

Thank you for this important work. I do apologize for the long wait on this review. As I communicated previously, I tried dozens of reviewers who did not respond to my request. I also did not feel like the one review secured was sufficient, so upon returning from my maternity leave, I wanted to ensure to give a careful review myself. Please attend to my comments as well as those from the reviewer.

Main comment:

Throughout the document, please pay attention and clarify to whenever “accept” and “acceptance” are used versus “uptake”. In many places (as I point out) it seems as if the authors equate acceptance to uptake, which eclipses the fact that low uptake may be influenced by access issues and systemic factors (that are independent of acceptance). A woman cannot accept a vaccine she is never offered, and it is unclear throughout the review if the factors and rates identified are within the context of women being offered the vaccine or in real-world situations where the vaccine may not be available. Some additional commentary about this issue (whether the main problem is acceptance or availability) would strengthen the discussion.

I also suggest making it more clear in Table 3 whether each factor identified relates to the outcome of acceptance, uptake or intention. These are all quite different and indicating the precise outcome would help the reader better interpret the ORs and RR reported in the right-hand column. This could be done by a symbol or an additional column.

Abstract:

- Please clarify what you mean by “barriers amongst health care providers” in the abstract. As it is written now, it is unclear if these are “barriers that inhibit healthcare providers from offering the vaccine to pregnant women”, or simply “barriers to vaccination as perceived by healthcare providers”.

- Your concluding statement identifies steps to improve “acceptance” of maternal influenza vaccine, suggesting that identifying barriers and facilitators to “acceptance” was the main goal of the study. However, this in not specified in the aim as written in the background. Barriers and facilitators can be multilevel, and include systemic barriers (supply, availability) that have nothing to do with acceptance. Please clarify in your aims and conclusion statement if your review was aimed at identifying barriers and facilitators specifically to acceptance or more broadly, to uptake.

Background

- Second sentence: Please clarify what population this hospitalization rate applies to (e.g. India?, low-and-middle income countries? Global?)

- Second paragraph: Beginning with the second sentence, please specify that the literature reviewed pertains to research globally (including high income countries) so that the reader does not misinterpret that this research is specific to LMICs

- Second paragraph, Last sentence: acceptability among who? Women? HCPs? Stakeholders? Also, why is it important to examine acceptability specifically versus other barriers? (see comment in abstract)

Methods

- “till” is not a full word. Please replace with “until” or “up until”

- Please capitalize “World Bank”

Results

- There are a few articles that are classified as having very high risk of bias. How did this work into your overall interpretation of results?

- Please capitalize Coast” (as in “Ivory Coast”)

- A more appropriate subheading would be “Vaccination rate” instead of “vaccination acceptance”. The term “acceptance” implies that all women were offered the vaccine and only some “accepted” and this might not be the case. If you do mean acceptance, a sentence is warranted about the background of the study explaining why this outcome is possible to measured (e.g. ensuring that all women in the denominator indeed had access to the vaccine).

- Similarly, “Women with higher education were 36-42% less likely to receive or accept influenza…” In this sentence, which do you mean, receive, or accept? They have quite different meanings.

- Similarly, I would not use the word “willing” as in “Women were more willing to receive” if non-availability is really a factor. Women could have been willing but unable to receive it because of non-availability. Using the word “willing” places all responsibility/blame on the woman when lack of uptake of the vaccine could be due to non-availability, independent of her willingness to receive it.

- In the first sentence of “Factors related to healthcare providers and health system” please specify, the odds of receiving the vaccine was higher when recommended by a HCP compared to what? When an HCP recommends against the vaccine? When someone other than the HCP recommends the vaccine? When an HCP offers no opinion on the vaccine? What was the comparison group here?

- Do you really mean “Lack of respect toward the HCP” or “Lack of respect by the HCP (toward the patient)?” Please clarify

Discussion

- Again, ensure and clarify that whenever the word “acceptance” is used, it means a vaccine was available to all women and they had the opportunity to accept

- I advise caution extrapolating the lower rate of vaccination in pregnancy vs. other population in reference 36 to vaccine hesitancy without sufficient supporting evidence. There may be other systemic factors that explain this lower rate.

- Does the evidence point to acceptance or access as the biggest barrier to vaccination during pregnancy in LMIC? Can you make a determination or comment on what the literature suggests, or point out the need for research to address this gap?

- “lack of research focusing policy makers” � I believe “on” is missing

- I am surprised that “improving health literacy” is the first point that you bring forth in your conclusion. I don’t believe that the review as a whole point to health literacy as the main issue. Please be more comprehensive in your conclusion to point out the myriad of multilevel factors that must be addressed to improve uptake (as well as acceptance).

- Thank you for pointing out the lack of research on this issue. I do hope your article inspires more research on this important topic.

Table 3

- Unless the number of antenatal visits reflects official recommendation/policy, I would put this factor under demographic/ clinical factors (not health systems)

Journal Requirements:

Reviewers' comments:

Reviewer's Responses to Questions

**Comments to the Author**

1. Is the manuscript technically sound, and do the data support the conclusions?

Reviewer #1: Yes

2. Has the statistical analysis been performed appropriately and rigorously? 

Reviewer #1: Yes

3. Have the authors made all data underlying the findings in their manuscript fully available?

Reviewer #1: Yes

4. Is the manuscript presented in an intelligible fashion and written in standard English?

Reviewer #1: Yes

5. Review Comments to the Author

Reviewer #1: This is a nicely written review on an important subject Determinants of maternal influenza vaccination in LMIC setting. This is an underexplored area, so this review is an important contribution. This review can be improved by including the reasons by rejecting large number of publications (including 19 reviews). Has this huge rejection introduced any bias, needs to be explained. If there is any way of analyzing these rejected 19 reviews and summarize findings from these reviews in a paragraph or table, it might be a way of minimizing impact of rejecting valid reviews.

I would recommend the authors to amend the manuscript accordingly so it becomes an important contribution from LMICs to global health.

6. PLOS authors have the option to publish the peer review history of their article (what does this mean?). If published, this will include your full peer review and any attached files.

Reviewer #1: No

---

## [Author Response · Author response to Decision Letter 0]

7 Oct 2021

Throughout the document, please pay attention and clarify to whenever “accept” and “acceptance” are used versus “uptake”. In many places (as I point out) it seems as if the authors equate acceptance to uptake, which eclipses the fact that low uptake may be influenced by access issues and systemic factors (that are independent of acceptance). A woman cannot accept a vaccine she is never offered, and it is unclear throughout the review if the factors and rates identified are within the context of women being offered the vaccine or in real-world situations where the vaccine may not be available. Some additional commentary about this issue (whether the main problem is acceptance or availability) would strengthen the discussion.

I also suggest making it more clear in Table 3 whether each factor identified relates to the outcome of acceptance, uptake or intention. These are all quite different and indicating the precise outcome would help the reader better interpret the ORs and RR reported in the right-hand column. This could be done by a symbol or an additional column.

Thank you very much for this constructive feedback. We have now used appropriate terminologies (acceptance or uptake) in the revised text. The table 3 is also revised accordingly. 

Abstract:

- Please clarify what you mean by “barriers amongst health care providers” in the abstract. As it is written now, it is unclear if these are “barriers that inhibit healthcare providers from offering the vaccine to pregnant women”, or simply “barriers to vaccination as perceived by healthcare providers”.

These are the barriers that inhibit the health care providers from offering the vaccine to pregnant women. This has been clarified in the abstract now. 

- Your concluding statement identifies steps to improve “acceptance” of maternal influenza vaccine, suggesting that identifying barriers and facilitators to “acceptance” was the main goal of the study. However, this in not specified in the aim as written in the background. Barriers and facilitators can be multilevel, and include systemic barriers (supply, availability) that have nothing to do with acceptance. Please clarify in your aims and conclusion statement if your review was aimed at identifying barriers and facilitators specifically to acceptance or more broadly, to uptake.

Thank you for this comment. As you have pointed out, we aim to review the uptake of maternal influenza vaccine broadly which includes acceptance wherever applicable. We have clarified this point in the revised aims and conclusions. 

Background

- Second sentence: Please clarify what population this hospitalization rate applies to (e.g. India?, low-and-middle income countries? Global?)

These are global hospitalisation rates for influenza related hospitalisation. We have revised the sentence as follows: 

“Globally, the influenza-related hospitalisation rate in pregnant women is estimated to be 42.1% (interquartile range (IQR), 22.5-60.4%) and among them around 8% (IQR, 5.9-12.7%) have severe disease that results in intensive care admission or death(4).”

- Second paragraph: Beginning with the second sentence, please specify that the literature reviewed pertains to research globally (including high income countries) so that the reader does not misinterpret that this research is specific to LMICs

The text pertains to global literature on maternal influenza. We have specified this in the revised text now. 

- Second paragraph, Last sentence: acceptability among who? Women? HCPs? Stakeholders? Also, why is it important to examine acceptability specifically versus other barriers? (see comment in abstract)

We have revised this sentence as follows: 

“This systematic review was planned to synthesize evidence about barriers and facilitators for maternal influenza vaccination amongst HCPs and pregnant women in LMICs.” 

Methods

- “till” is not a full word. Please replace with “until” or “up until”

We have revised the sentence as follows: 

“We included studies published until Dec 31, 2020 amongst women in the antenatal as well as postnatal period.”

- Please capitalize “World Bank”

The text has been revised accordingly. 

Results

- There are a few articles that are classified as having very high risk of bias. How did this work into your overall interpretation of results?

Thank you. We have addressed this in the revised text in discussion: 

“Four out of 11 studies had high or very high risk of bias. This is one of the serious limitations of this review which reduces the confidence in the results from these studies.” 

- Please capitalize Coast” (as in “Ivory Coast”)

The text has been revised accordingly. 

- A more appropriate subheading would be “Vaccination rate” instead of “vaccination acceptance”. The term “acceptance” implies that all women were offered the vaccine and only some “accepted” and this might not be the case. If you do mean acceptance, a sentence is warranted about the background of the study explaining why this outcome is possible to measured (e.g. ensuring that all women in the denominator indeed had access to the vaccine).

We would like to clarify that ref 25, 27 and 28 report vaccination rate or uptake and ref 23, 24, 29 report willingness to get vaccinated if the vaccine is made available. To include both these scenarios, we have changed the heading to “Vaccine acceptance or vaccine uptake” in the revised text. 

- Similarly, “Women with higher education were 36-42% less likely to receive or accept influenza…” In this sentence, which do you mean, receive, or accept? They have quite different meanings.

The verb accept would be more appropriate in this case, as the influenza vaccination was provided under government program in Peru and was available to all free of cost. We have made the required change. 

- Similarly, I would not use the word “willing” as in “Women were more willing to receive” if non-availability is really a factor. Women could have been willing but unable to receive it because of non-availability. Using the word “willing” places all responsibility/blame on the woman when lack of uptake of the vaccine could be due to non-availability, independent of her willingness to receive it.

The term willingness has been used for three studies reported from Pakistan (not in immunisation program but recommended to pregnant women), Ivory coast (vaccine not free but responses of the participants were recorded for their intent to get vaccinated) and Gambia (vaccine was under clinical trial mode and participants were asked about their willingness / intent for use in pregnancy if made available). As these studies have investigated the willingness or intent to get vaccinated, the word ‘willingness’ is appropriate here. 

- In the first sentence of “Factors related to healthcare providers and health system” please specify, the odds of receiving the vaccine was higher when recommended by a HCP compared to what? When an HCP recommends against the vaccine? When someone other than the HCP recommends the vaccine? When an HCP offers no opinion on the vaccine? What was the comparison group here?

The comparison group here was women who did not receive any recommendation from their health care providers. We have modified the sentence as follows: 

“The odds of receiving influenza vaccine were reported 2.5 to 74 times higher in pregnant women who received a recommendation from an HCP as compared to those who did not receive any recommendation (30)(27)(26).”

- Do you really mean “Lack of respect toward the HCP” or “Lack of respect by the HCP (toward the patient)?” Please clarify.

Thank you for this comment. We have clarified the term as “lack of respect by HCPs towards confidentiality of the participants” in the revised text. 

Discussion

- Again, ensure and clarify that whenever the word “acceptance” is used, it means a vaccine was available to all women and they had the opportunity to accept. 

Thank you. We have now used the word acceptance in the discussion only where it is appropriate. 

- I advise caution extrapolating the lower rate of vaccination in pregnancy vs. other population in reference 36 to vaccine hesitancy without sufficient supporting evidence. There may be other systemic factors that explain this lower rate.

We have revised the text as follows: 

“However, the coverage for influenza vaccination was lower during pregnancy than other populations (e.g. children and elderly) in Peru and Nicaragua(36). These findings are similar to other global data indicating low uptake of influenza vaccine uptake during pregnancy(37)”

- Does the evidence point to acceptance or access as the biggest barrier to vaccination during pregnancy in LMIC? Can you make a determination or comment on what the literature suggests, or point out the need for research to address this gap?

Our evidence from Peru study points out that acceptance may play equally important role. To address this, we have added the following sentence in the discussion:

“However, as pointed out in the Peru study (25), the vaccine uptake may remain low despite subsidized vaccine program. This highlights the importance of including awareness campaigns for general public and health care providers in order to improve vaccine coverage.”

However, we have a limitation that only three of the included studies had vaccination included in the health programs and the other studies only assessed willingness to get vaccination in the absence of actual program. Hence, we have added following line as a limitation in the discussion: 

“Except Nicaragua (27) (28) and Peru (25), none of the countries had maternal influenza vaccination included in the health program. Hence, there was limited information available about the logistic or operational factors related to availability of influenza vaccine for pregnant women in these countries, which can play very important role in the success of implementation. 

“lack of research focusing on the policy makers” � I believe “on” is missing

Thank you for pointing out this typographical error. We have corrected the mistake in the revised text. 

I am surprised that “improving health literacy” is the first point that you bring forth in your conclusion. I don’t believe that the review as a whole point to health literacy as the main issue. Please be more comprehensive in your conclusion to point out the myriad of multilevel factors that must be addressed to improve uptake (as well as acceptance).

Thank you for your feedback. The conclusion is modified accordingly. The revised conclusion is as follows: 

“Higher educational status, better access to antenatal care, perceived risk of influenza during pregnancy, perceived benefits of influenza vaccination during pregnancy, recommendation by health care providers and inclusion of maternal influenza vaccine in health policy were important facilitators for maternal influenza vaccine uptake in LMIC. Fear of adverse effects, uncertainty about the benefits of vaccination and ineffective health communication regarding the influenza vaccine were barriers identified. 

Thus, while availability of influenza vaccine in public health system can be a key to the success of vaccine implementation program, increasing the public health awareness about need and benefits of maternal influenza vaccination amongst pregnant women as well as their health care providers is crucial to improve the acceptance of maternal influenza vaccination in low and middle-income countries.” 

Thank you for pointing out the lack of research on this issue. I do hope your article inspires more research on this important topic.

Thank you for your feedback. We do hope the same. 

Table 3

- Unless the number of antenatal visits reflects official recommendation/policy, I would put this factor under demographic/ clinical factors (not health systems). 

The minimum number of antenatal visits at both public and private health care setup for a healthy pregnancy is decided by a health program in the country. Hence, we argue that number of antenatal visits should be part of health system.

---

## [Decision Letter · Decision Letter 1]

3 Nov 2021

PONE-D-21-08838R1Determinants of maternal influenza vaccination in the context of low- and middle-income countries: A systematic reviewPLOS ONE

Dear Dr. Apte,

Thank you for the careful response to the comments from myself and the reviewer. The reviewer has recommended the manuscript be accepted, and I agree with this recommendation. The manuscript is very much improved and will make a good contribution to PLOS ONE. However, before the manuscript is accepted and sent for proofing, I wanted to give you the opportunity to address a few small items from Reviewer 1 as well as the following suggestions from me regarding Table 3.

We look forward to receiving your revised manuscript.

Kind regards,

Emily A Hurley, M.P.H., Ph.D.

Academic Editor

PLOS ONE

Journal Requirements:

Additional Editor Comments (if provided):

- The addition of the final column in Table 3 is helpful in understanding the outcome of each study. In the "Vaccine acceptance and vaccine uptake" section of the manuscript, before you introduce Table 3, it might be helpful to orient the reader to this column these terms by defining them. For example, you might say "Outcomes in the studies included overall vaccine uptake as well as vaccine acceptance (patient agreement to vaccine assuming it was offered) (Table 3)."

- "uptake" might also be added to the title of table 3 ".. uptake and acceptance of..."

- Please clarify what you mean by "clubbing". Do you mean "combining"?

- I agree that variable of number of antenatal visits, if it purely reflects a policy, should remain under health systems factors. Would be appropriate to label this variable "Recommended number of antenatal visits" so that the reader knows this reflects the ideal number as set by national recommendations? If not, it might still fit better under "clinical factors". Similar to the variable "vaccination in third trimester", number of antenatal visits is determined by both the individual and the health system (as even if 4 visits are recommended, many women will not complete that amount).

Reviewers' comments:

Reviewer's Responses to Questions

**Comments to the Author**

1. If the authors have adequately addressed your comments raised in a previous round of review and you feel that this manuscript is now acceptable for publication, you may indicate that here to bypass the “Comments to the Author” section, enter your conflict of interest statement in the “Confidential to Editor” section, and submit your "Accept" recommendation.

Reviewer #1: All comments have been addressed

2. Is the manuscript technically sound, and do the data support the conclusions?

Reviewer #1: Yes

3. Has the statistical analysis been performed appropriately and rigorously? 

Reviewer #1: Yes

4. Have the authors made all data underlying the findings in their manuscript fully available?

Reviewer #1: Yes

5. Is the manuscript presented in an intelligible fashion and written in standard English?

Reviewer #1: Yes

6. Review Comments to the Author

Reviewer #1: The revised manuscript has addressed most of the issues raised by reviewers.

The suggestion "This review can be improved by including the reasons by rejecting large number of publications (including 19 reviews). Has this huge rejection introduced any bias, needs to be explained. If there is any way of analyzing these rejected 19 reviews and summarize findings from these reviews in a paragraph or table, it might be a way of minimizing impact of rejecting valid reviews" has not been addressed adequately though.

The reference numbers 32 and 34 are repeat, need to check and revise the numbers.

The modified manuscript can be accepted as it's a valuable contribution from LMICs to the field.

7. PLOS authors have the option to publish the peer review history of their article (what does this mean?). If published, this will include your full peer review and any attached files.

Reviewer #1: No

---

## [Author Response · Author response to Decision Letter 1]

24 Nov 2021

Additional Editor Comments (if provided):

- The addition of the final column in Table 3 is helpful in understanding the outcome of each study. In the "Vaccine acceptance and vaccine uptake" section of the manuscript, before you introduce Table 3, it might be helpful to orient the reader to this column these terms by defining them. For example, you might say "Outcomes in the studies included overall vaccine uptake as well as vaccine acceptance (patient agreement to vaccine assuming it was offered) (Table 3)."

Thank you for this valuable input. We have added this sentence under vaccine acceptance and vaccine uptake. 

- "uptake" might also be added to the title of table 3 ".. uptake and acceptance of..."

Thank you. We have made the required changes. 

- Please clarify what you mean by "clubbing". Do you mean "combining"?

We clarified this by replacing the term clubbing with combining. 

- I agree that variable of number of antenatal visits, if it purely reflects a policy, should remain under health systems factors. Would be appropriate to label this variable "Recommended number of antenatal visits" so that the reader knows this reflects the ideal number as set by national recommendations? If not, it might still fit better under "clinical factors". Similar to the variable "vaccination in third trimester", number of antenatal visits is determined by both the individual and the health system (as even if 4 visits are recommended, many women will not complete that amount).

Agree with this. We have added antenatal visits under clinical factors. 

Reviewers' comments:

Reviewer's Responses to Questions

Comments to the Author

1. If the authors have adequately addressed your comments raised in a previous round of review and you feel that this manuscript is now acceptable for publication, you may indicate that here to bypass the “Comments to the Author” section, enter your conflict of interest statement in the “Confidential to Editor” section, and submit your "Accept" recommendation.

Reviewer #1: All comments have been addressed

2. Is the manuscript technically sound, and do the data support the conclusions?

Reviewer #1: Yes

3. Has the statistical analysis been performed appropriately and rigorously?

Reviewer #1: Yes

4. Have the authors made all data underlying the findings in their manuscript fully available?

Reviewer #1: Yes

5. Is the manuscript presented in an intelligible fashion and written in standard English?

Reviewer #1: Yes

6. Review Comments to the Author

Reviewer #1: The revised manuscript has addressed most of the issues raised by reviewers.

The suggestion "This review can be improved by including the reasons by rejecting large number of publications (including 19 reviews). Has this huge rejection introduced any bias, needs to be explained. If there is any way of analyzing these rejected 19 reviews and summarize findings from these reviews in a paragraph or table, it might be a way of minimizing impact of rejecting valid reviews" has not been addressed adequately though.

We have added the following text at the end of the discussion: 

“We have excluded full text articles from high income countries and focused on the studies from LMIC only. This is because the problems faced by LMIC are likely to be different from higher income countries due to differences in the socioeconomic status, literacy and health care access for pregnant women. However, we have discussed the findings from these studies at relevant places in the introduction and discussion.”

We have now included all the relevant reviews (listed below) from our excluded list of articles in the background discussion. 

1. Hirve S, Lambach P, Paget J, Vandemaele K, Fitzner J, Zhang W. Seasonal influenza vaccine policy, use and effectiveness in the tropics and subtropics - a systematic literature review. Influenza and other respiratory viruses. 2016 Jul;10(4):254–67

2. Regan A, Haberg S, Fell DB. Current Perspectives on Maternal Influenza Immunization. Current Tropical Medicine Reports. 2019;6(4):239–49.

3. Lutz CS, Carr W, Cohn A, Rodriguez L. Understanding barriers and predictors of maternal immunization: Identifying gaps through an exploratory literature review. Vaccine. 2018 Nov;36(49):7445–55.

4. Morales KF, Menning L, Lambach P. The faces of influenza vaccine recommendation: A Literature review of the determinants and barriers to health providers’ recommendation of influenza vaccine in pregnancy. Vaccine. 2020 Jun;38(31):4805–15.

5. Buchy P, Badur S, Kassianos G, Preiss S, Tam JS, P. B, et al. Vaccinating pregnant women against influenza needs to be a priority for all countries: An expert commentary. International Journal of Infectious Diseases 2020 Mar;92:1–12.

6. Yuen CYS, Tarrant M. Determinants of uptake of influenza vaccination among pregnant women - a systematic review. Vaccine. 2014 Aug;32(36):4602–13.

7. Yuen CYS, Tarrant M. A comprehensive review of influenza and influenza vaccination during pregnancy. The Journal of perinatal & neonatal nursing. 2014;28(4):261–70. 

8. Wong VWY, Lok KYW, Tarrant M. Interventions to increase the uptake of seasonal influenza vaccination among pregnant women: A systematic review. Vaccine. 2016 Jan;34(1):20–32.

9. Raya BA, Edwards KM, Scheifele DW, Halperin SA, B. AR, K.M. E, et al. Pertussis and influenza immunisation during pregnancy: a landscape review. The Lancet Infectious Diseases. 2017 Jul;17(7):e209–22. 

10. Phadke VK, Omer SB. Maternal vaccination for the prevention of influenza: current status and hopes for the future. Expert Review of Vaccines 2016 Oct;15(10):1255–80

The reference numbers 32 and 34 are repeat, need to check and revise the numbers.

Thank you for pointing this out. We have removed the duplicate reference. 

The modified manuscript can be accepted as it's a valuable contribution from LMICs to the field.

7. PLOS authors have the option to publish the peer review history of their article (what does this mean?). If published, this will include your full peer review and any attached files.

Do you want your identity to be public for this peer review? For information about this choice, including consent withdrawal, please see our Privacy Policy.

Reviewer #1: No

---

## [Editor Report · Decision Letter 2]

7 Jan 2022

Determinants of maternal influenza vaccination in the context of low- and middle-income countries: A systematic review

PONE-D-21-08838R2

Dear Dr. Apte,

We’re pleased to inform you that your manuscript has been judged scientifically suitable for publication and will be formally accepted for publication once it meets all outstanding technical requirements.

Kind regards,

Emily A Hurley, M.P.H., Ph.D.

Academic Editor

PLOS ONE
---

## [Editor Report · Acceptance letter]

17 Jan 2022

PONE-D-21-08838R2 

Determinants of maternal influenza vaccination in the context of low- and middle-income countries: A systematic review 

Dear Dr. Apte:

I'm pleased to inform you that your manuscript has been deemed suitable for publication in PLOS ONE. Congratulations! Your manuscript is now with our production department. 

Kind regards, 

on behalf of

Dr. Emily A Hurley 

Academic Editor

PLOS ONE